# Using 3D Microscope for Hepatic Artery Reconstruction in Living Donor Liver Transplant

**DOI:** 10.3390/jcm11206195

**Published:** 2022-10-20

**Authors:** Ching-Min Lin, Shih-Lung Lin, Yu-Ju Hung, Chih-Jan Ko, Chia-En Hsieh, Yao-Li Chen, Chien-Hsiang Chang

**Affiliations:** 1Department of Surgery, Changhua Christian Hospital, Changhua 500209, Taiwan; 2Department of Plastic, Reconstructive, and Hand Surgery, Changhua Christian Hospital, Changhua 500209, Taiwan; 3Organ Transplant Center, Changhua Christian Hospital, Changhua 500209, Taiwan; 4Department of Surgery, Liver Transplant Center, Chung Shan Medical University Hospital, Taichung 402306, Taiwan; 5Department of Nursing, Liver Transplant Center, Chung Shan Medical University Hospital, Taichung 402306, Taiwan

**Keywords:** 3D digital microscope, exoscope, exoscopic anastomosis, artery anastomosis, liver transplant

## Abstract

Introduction: This study compares the intraoperative process of hepatic artery anastomosis using conventional microscope and novel 3D digital microscope and discusses our technique and operative set-up. Method: A retrospective comparative cohort study with 46 hepatic artery reconstructions in living donor liver transplant patients. Either an operational microscope (control group) or a 3D digital microscope Mitaka Kestrel View II (study group) was used for hepatic artery anastomosis. We then discuss and share our institution’s experience of improving surgical training. Results: Both operation instruments provide effective and comparable results. There was no statistical difference regarding operational objective results between conventional microscope and exoscope. Both instruments have no hepatic artery size limit, and both resulted in complete vessel patency rate. Conclusions: There was no statistical differences regarding hepatic artery anastomosis between microscope and exoscope cohorts. Microsurgeons should perform hepatic artery anastomosis efficiently with the instruments they are most proficient with. Yet, exoscope provided better ergonomics in the operation room and lessened musculoskeletal strain, allowing surgeons to work in a more neutral and comfortable posture while allowing the first assistant to learn and assist more effectively. Using exoscope with micro-forceps and modified tie technique make artery reconstruction easier.

## 1. Introduction

Living donor liver transplant (LDLT) is considered the best treatment option for advanced liver disease. The most crucial and complex step in LDLT is the reconstruction of hepatic artery. The early experience of LDLT by using surgical loupe resulted in high rates of hepatic artery thrombosis, hepatic necrosis, graft loss and mortality up to 50% [1,2]. Patients who develop hepatic artery thrombosis have decreased survival even if they undergo re-transplantation [3]. Since the introduction of microvascular anastomosis, reported incidence of hepatic artery reconstruction complication rates was dramatically reduced to between 0 and 6% [4]. Hence, a shift to microscopic anastomosis was made. The shift from surgical loupe to microscope can have long learning curves due to differences in the entire surgical equipment and technology. Microvascular reconstruction is the trend in most hospitals, so here we share our experience with 3D digital microscopy. The 3D exoscope is a novel high-definition digital camera. Its conversion of digital data allows us to have stereovision. Since it does not rely on the eyepiece, surgeons can have a wider range of motion. Its operation method is similar to that of conventional microscopes, and thus the learning curve, after switching from conventional microscopy to exoscope, is shortened. This retrospective study compared a single microsurgeon’s intraoperative process on hepatic artery anastomosis during living donor liver transplants using operational microscopy versus 3D digital microscopy and discusses the benefit of newly emerging exoscopic technology. 

## 2. Materials and Methods

We conduct a retrospective study of hepatic artery reconstruction in living donor liver transplant patients between April 2018 to April 2019 at Changhua Christian Hospital. All hepatic artery anastomosis was performed by a single plastic surgeon Dr. Shih-Lung Lin. A total of 46 charts were examined: 23 patients received microsurgery via operational microscopy (Control group) and 23 patients received microsurgery via 3D digital microscopy Mitaka Kestrel View II (Mitaka Kohki Co., Ltd., Tokyo, Japan) (Study group). Collected data included patient and donor’s demographic data, medical history, tissue ischemia time and procedural characteristics. Surgical outcome as hepatic artery perfusion was evaluated via Doppler ultrasound immediately post anastomosis and before reversal of general anesthesia at the end of the operation. Descriptive statistics and categorical variables were tabulated based on medical record. All analyses were performed via SPSS Statistic, version 23.0 (SPSS Inc., Chicago, IL, USA). A 2-tail *p*-value < 0.05 was considered statistically significant. The study was conducted in accordance with the Declaration of Helsinki and approved by the institutional review board and ethics committee of Changhua Christian Hospital (CCH IRB No. 181020), which waived the need for patient consent.

## 3. Results

The study included 46 patients who underwent hepatic artery reconstruction as part of a living donor liver transplant. Overall, gender was distributed equally between control and study group, with each group having 16 males in 23 patients. There was no statistical difference between recipient age and donor age in both groups. Both the control and study group had two patients (8.7%) with hypertension. The control group had 8 patients (34.8%) with diabetes mellitus and 12 smokers (52.2%), whereas the study group had 5 patients (21.7%) with diabetes mellitus and 15 smokers (65.2%). These demographic data did not provide any clinical significance. In the control group, the recipient artery size averaged 3.65 ± 0.647 mm and donor artery size averaged 2.43 ± 0.662 mm. In the study group, the recipient artery size averaged 3.91 ± 0.900 mm and donor artery size averaged 2.70 ± 0.765 mm. Intraoperative variables, including average cold ischemia time, warm ischemia time, anastomosis time, and blood loss, were of no statistical difference between the two comparative cohorts. 

## 4. Discussion

This retrospective cohort study investigates the usage and benefits of 3D digital microscope in hepatic artery reconstruction during living donor liver transplant. Although operative techniques for hepatic transplant have matured enough to yield high success rates, we propose a few modifications that can be adopted to make vascular anastomosis in hepatic transplant easier for young surgeons. 

### 4.1. The Use of Micro-Forceps

Forceps are commonly used during operations. Each surgical specialty has its own custom forceps that is designed to suit its operative approaches (Figure 1). The 15 cm long forceps (Figure 1a) is a standard instrument in most general surgery operations. Yet, when used in hepatic artery anastomosis, it is not effective. As seen in Figure 2, the end of a 15 cm long forceps, when held in the hand of surgeon during artery anastomosis, is against the operator’s palm, and the operator’s fingers are pinched to have effective grip. This posture is tiresome for the hand and provides little lever working space. Furthermore, the surgeon’s hand motion is often limited by the ribcage encasing the liver during liver transplantation surgery. To overcome this limitation, we use 25 cm or 20 cm long micro-forceps for the procedure (Figure 1b). As seen in Figure 2, the extended length of this micro-forceps allows surgeons’ hand to work in a more relaxed grip with a working field that is wider and deeper. Leverage in longer forceps allow precise movement at the forceps’ tip without much hand movement; this is essential for artery reconstruction. When both the primary surgeon and assistant surgeon use micro-forceps during the procedure (Figure 2c), the assistant is able to provide more precise assistance without blocking the surgeon’s working field, so that the anastomosis can proceed smoothly and efficiently.

### 4.2. Tying Surgical Knots

Surgical residents were conventionally taught to hold the needle-holder with their dominant hand, usually the right-hand, and have the forceps in their non-dominant hand, usually the left-hand. Surgical knots were made by circling the forceps around the needle holder. However, during hepatic artery anastomosis, the ribcage encasing the liver will limit the surgeon’s field of exercise. Since the working field is vertically beneath the hand, as seen in Figure 2, surgical knots are often loose due to inadequate tie. We propose circling the needle-holder around the still forceps. The right-hand-held needle-holder has more working space than the left-hand forceps. Firmer surgical knots can be made this way than the conventional method.

### 4.3. The Operation Setup: Operative Microscope

Figure 3 shows the operation setup with conventional operative microscope (control group). The main operator is on the left side and assistant operator on the right side of the microscope. To accommodate the microscope, operators often maintain a forward-leaning, stiff-neck and shrugged-shoulder position. If arm-board is used, operators may need to stand with twisted trunk position and slanted shoulder to gain visualization to the microscopic field. In addition, with only one objective lens and the microscope usually being set up to suit the primary surgeon, the assistant surgeon often stands in awkward non-ergonomic positions, as seen in Figure 3.

### 4.4. The Operation Setup: Exoscope

Figure 4 shows the operative setup with 3D digital microscope Mitaka Kestrel View II (Mitaka Kohki Co., Ltd., Tokyo, Japan) (study group). As seen in Figure 4, both operators are standing in an upright position with a relaxed shoulder. A real time projection of the operation field allows all bystanders, such as the scrub nurse, anesthesiologist and surgical residents, to learn intraoperative techniques and follow the progress of the operation. In addition, with the advancement in technology, we have an aging surgeon population. Looking at a projection screen at a distance is easier on the eyes for surgeons with presbyopia. With less physical discomfort, surgeons will have better concentration and will feel more comfortable to teach and learn throughout this complicated surgery. 

### 4.5. Physical Benefits of Our Approach

Surgeons endure long and strenuous training years with high level of stress to perfect their surgical techniques in order to improve patients’ surgical outcome and prognosis. This pursuit of perfection comes at the cost of surgeons’ physical and mental health. Studies have found that 60–90% of all surgeons experience painful musculoskeletal conditions in their neck, back or shoulders [5,6,7]. Microsurgeons experience these symptoms during or after microscope use [8]. Our approach with using micro-forceps stabilizes the hand during vascular reconstruction with more precise movement and the modified suture tying method provides a larger working field with firmer ties made. The use of exoscopes, such as 3D microscopy, can neutralize mal-posture and alleviate work-related injuries. However, surgeons cannot benefit themselves at the cost of jeopardizing patient’s health. 

Rosenblatt et al. analyze surgeons’ intraoperative posture and found the three most common malpositions that contributed to musculoskeletal injuries: forward head position, improper shoulder elevation and internal rotations, and pelvic girdle asymmetry [9]. As shown in Figure 3, the forward-leaning, stiff-neck and shrugged-shoulder positions cause fatigue in deltoid and trapezius muscles. Due to obstacles, surgeons need to maintain a twisted position to gain visualization of the operation field. This sustained twisting causes an asymmetrical loading on back and leg muscles, leading to work-related musculoskeletal pain. Other risk factors that led to increased muscle activity and muscle fatigue include prolonged static posture, hyperflexion of cervical spine and back-bent posture [6,10,11]. Hence, operating with a more neutral position should be reminded and reinforced among surgeons.

### 4.6. Operative Benefits of Our Approach

Although Pafitanis et al. found that exoscopic microvascular anastomosis was more time consuming than conventional methods [12], our comparative analysis of hepatic artery reconstruction was performed via operational microscopy and via 3D digital microscopy, showing no difference in operative outcome (Table 1). Both instruments have no hepatic artery size limit, and both have complete vessel patency rate. This non-inferiority is in concordance with a previous study [8]. 

Modern digital exoscopes have their forte with surgeons. Its application ranges from open surgery to microsurgery and its ergonomics design, the projection of microscopic image onto a 3D monitor, allow surgeons to operate in a heads-up position, alleviating cervical musculoskeletal fatigue and work-related injuries. For surgeons with presbyopia, looking at a distant screen is more comfortable than a close-up image. Moreover, this real-time projection allows all active participants in the operation room to see the progress of the operation and allow bystanding surgical residents to learn intraoperative skills. The 3D digital microscopy is an asset to surgeons in non-reconstructive specialties because of its similar operative method with laparoscopy, and the switch from microscope to exoscope with good quality outcome can be achieved in a relatively short practice time [13].

We acknowledge that our study had limitations and bias. Our study sample size is not sufficient to provide comparative long-term follow-up results to assure that the two types of arterial anastomosis are equivalent in terms of complications. Furthermore, the retrospective nature of this study did not allow us to quantify ergonomic parameters via questionnaires or provide objective measurements. Although intuitive, a prospective study with larger sample size in the future can offer better insight.

## 5. Conclusions

Vascular anastomosis is the foundation of plastics and reconstructive surgery, and technological advancements in this field have revolutionized the approach to reconstructive microsurgery. The 3D microscopy is a novel exoscope that is at least equivalent to operating microscopy in an intraoperative procedure. When microvascular anastomosis is performed by experts using 3D microscopy there is no negative clinical impact to patients and it offers improved ergonomics, image quality and accessibility to the surgical field. New technologies have their advantages and as surgical residents learn to use these instruments and techniques, comparative analysis of surgical outcome for patients is just as important as operator’s overall comfort during the operation. 

## Figures and Tables

**Figure 1 jcm-11-06195-f001:**
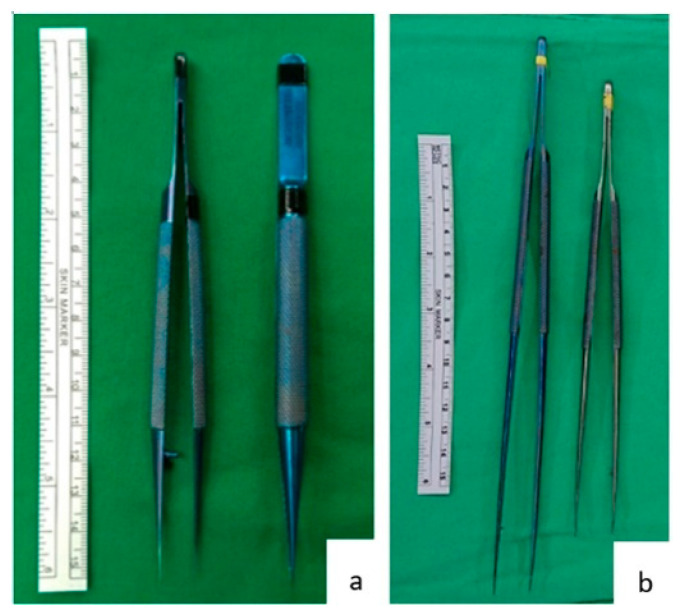
Forceps and micro-forceps. (**a**) A 15 cm long tissue forceps, a standard instrument in most general operations. (**b**) A 25 cm long and a 20 cm long micro-forceps used during hepatic artery anastomosis at our institution.

**Figure 2 jcm-11-06195-f002:**
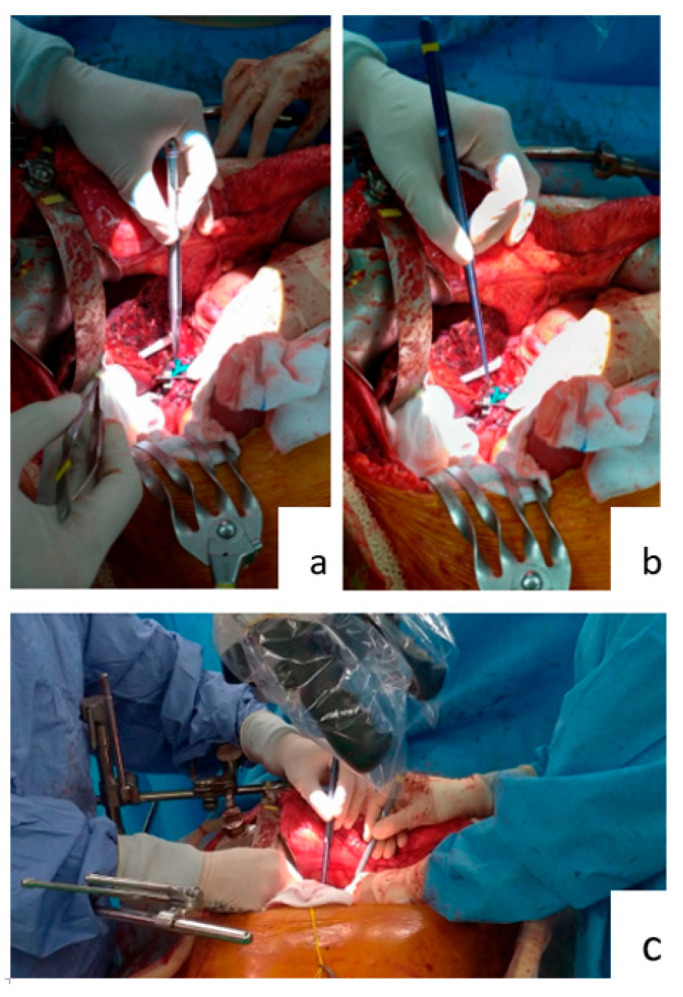
Working with forceps and micro-forceps. (**a**) Surgeon using the standard 15 cm long forceps. (**b**) Surgeon using the 25 cm long micro-forceps. (**c**) Surgeon and first assistant both using micro-forceps.

**Figure 3 jcm-11-06195-f003:**
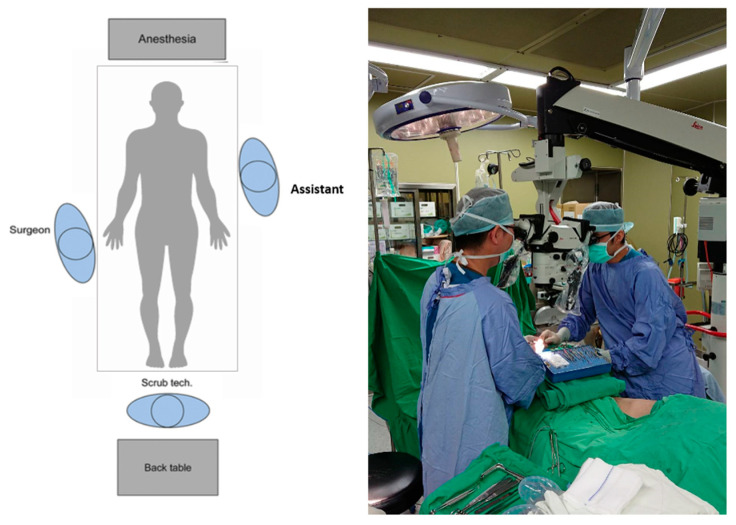
Operative setup for conventional microscope.

**Figure 4 jcm-11-06195-f004:**
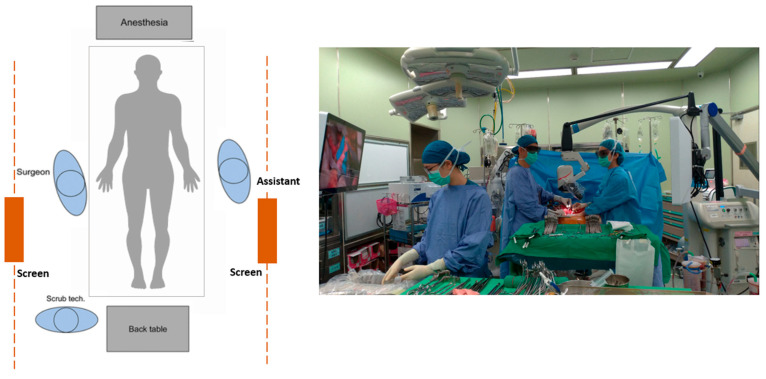
Operative setup for 3D digital microscope.

**Table 1 jcm-11-06195-t001:** Demographic and intraoperative features between conventional microscope (control group) and exoscope (study group).

Demographic and Clinical Features	Control Group	Study Group		Total
	*n* = 23	*n* = 23	*p*	*n* = 46
	Mean ± SD	Mean ± SD		Mean ± SD
	(range)	(range)		(range)
Recipient age (years)	54.78 ± 7.292	53.78 ± 8.339	0.575	54.28 ± 7.762
	(35–67)	(39–70)		(35–70)
Donor age (years)	32.30 ± 9.152	32.22 ± 11.342	0.575	18.0 ± 54.0
	(19–54)	(18–52)		(32.26–10.190)
Recipient artery size (mm)	3.65 ± 0.647	3.91 ± 0.900	0.204	2.78 ± 0.786
	(2.0–5.0)	(2.0–6.0)		(2.0–6.0)
Donor artery size (mm)	2.43 ± 0.662	2.70 ± 0.765	0.216	2.57 ± 0.720
	(2.00–4.00)	(2.0–4.0)		(2.0–4.0)
Anastomosis time (min)	25.22 ± 7.373	23.91 ± 5.316	0.422	24.57 ± 6.390
	(12.0–38.0)	(15.0–35.0)		(12.0–38.0)
Blood loss (mL)	2985.65 ± 2021.583	4854.35 ± 7992.623	0.391	3875.00 ± 5848.901
	(800.0–1000.00)	(500.0–40,000.0)		(500.0–40,000.0)
Cold ischemia time (min)	42.00 ± 22.817	49.70 ± 21.231	0.059	45.85 ± 22.137
	(13.0–128.0)	(25.0–130.0)		(13.0–130.0)
Warm ischemia time (min)	19.83 ± 5.589	21.35 ± 6.050	0.208	20.59 ± 5.810
	(13.0–33.0)	(13.0–35.0)		(13.0–35.0)
	(%)	(%)		(%)
Gender (male)	16 (69.6)	16 (69.6)	1.000	(32) 69.6
Diabetes mellitus	8 (34.8)	5 (21.7)	0.326	13 (28.3)
Hypertension	2 (8.7)	2 (8.7)	1.000	4 (8.7)
Smoking	12 (52.2)	15 (65.2)	0.369	27 (58.7)

## Data Availability

All data are available upon reasonable request.

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
