# Peer review of "Using 3D Microscope for Hepatic Artery Reconstruction in Living Donor Liver Transplant"

_jcm, 2022, doi:10.3390/jcm11206195_

Round 1
Reviewer 1 Report
This is an exciting manuscript. However, I wonder if this could be considered a full research article or short communication. Nevertheless, anything depicting new technologies or advancements in healthcare is always encouraged.
It seems multiple authors write the manuscript. Therefore, many sentences need to be revised. It is advised to revise the manuscript to improve clarity and grammar thoroughly.
For example, some sentences, however, are not limited to...
A retrospective cohort study of hepatic artery reconstruction as part of living donor liver transplant patients between 2018 April and 2019 April done by single plastics surgeon Dr. Shih-Lung Lin at medical center Changhua Christian Hospital was conducted.
There was no statistical significance between recipient age and donor age in both groups.
Intraoperative variables, including average cold ischemia time, warm ischemia time, anastomosis time, and blood loss, were not statistically significant differences in favorable hepatic artery anastomosis between the two comparative cohorts.
Besides, authors should number the pictures like a, b, c, and mark left and right images.
I think the reference should be as per the journal format.
Author Response
We thank the reviewer for the time and effort invested into reviewing our manuscript. Here are our response to your comments and feedback.
- We have revised and edited our manuscript to improve clarity and grammar thoroughly.
- We have number the pictures with a,b,c for reference clarity.
- We have revised and edited our reference style according to ACS reference style.
Reviewer 2 Report
In this article, the Authors discussed the benefit of a 3D microscope in liver transplant (LT) operation compared with the traditional operational microscope. Authors found 3D microscope provided better ergonomics and less musculoskeletal pain, stiff neck, and shrugged shoulder position to the surgeons, giving the surgeon a relaxed and comfortable posture during operation, and allowing the first assistant to learn and help more effectively. Meanwhile, the authors found 3D microscope didn’t increase hepatic artery anastomosis efficiency. On the other hand, authors suggested replacing normal micro-forceps with long forceps during operation. Pictures and tables have been used to prove their results. It is an interesting article comparing the new 3D microscope with the traditional operational microscope; however, I still have some concerns regarding this article.
1. Why did not the company replace the traditional operational eye lens with the small monitors just in front of surgeons, because it is difficult to see the faraway huge monitor, which is easy to be covered by others?
2. Authors still need to compare the hepatic artery thrombosis, hepatic necrosis, graft loss, and mortality after LT in the table.
Author Response
We thank the reviewer for the time and effort invested into reviewing our manuscript. Here are our response to your comments and feedback:
Point 1. Conventional operational eye lens works via light and optics. To achieve magnification via optical microscope, it must have a large-enough light-receiving lens. For example, monocular camera need longer focus lens to magnify distant objects. To convert optic lens image and project it onto a screen must require digital imaging. Small screen will sacrifice imaging quality. Hence, traditional operational eye lens cannot project image onto a small monitors and place it in front of the surgeon. Exosopes, such as Mitaka Kestrel View II, is a high-definition digital imaging system that enables surgeons to see a magnified, three-dimensional image of the surgical field during microsurgeries. With advancement in technology, we have a population of aging-surgeons. Elder surgeons, especially ones with presbyopia, will find a distant larger screen more comfortable to watch than a close-up small screen.
Point 2. Thank you for this helpful suggestion. We thought about incorporating these variables into our table yet due to our small sample size and having complete 100% vessel patency rate in both cohorts, we did not feel that it would not reflect true complication rates. A retrospective study with larger sample size in the future will offer better insight. Perhaps we should modify our title to “How to use 3D microscopy for hepatic artery reconstruction in living donor liver transplant” to better present our paper content?
Reviewer 3 Report
Congratulations on the excellent results and zero thrombosis rate after hepatic artery anastomosis using the operating microsope and exoscope
In experienced hands, both techniques seem to have equal results; the main advantage of the 3D microscope seem to be better optics, better ergonomics to the operating team and teaching value. To compare these somewhat intangible but important advantages, my suggestion would be to have a prospective study in future to quantify these parameters and compare the differences
Congratulations on very good results and 100% hepatic artery patency rates with both conventional microscope and the 3D exoscope, a testimony to the importance of expertise.
Major points
The manuscript seems to be a mix of 1) a review of technical tips by the authors and 2) a comparison of certain objective parameters like hepatic artery size, patency rates, operating time and less easily quantifiable parameters like ergonomics, teaching value between the two cohorts. A clear distinction should be made between rigorous results and expert opinion
Since this is a retrospective study, how were parameters like musculoskeletal fatigue, ergonomics accurately compared? While it may be certainly true that the 3D digital microscope scores over the conventional operating microscope in this regard, this study design does not permit this comparison.
Similarly better vison/optics and teaching value are common sense observations made by the authors. However, there are no quantifiable parameters in this study that permit these conclusions to be drawn
Technical tips provided in the manuscript discussion on use of long forceps, knot tying and posture of the surgeons in the operating room setup are extremely useful. However, these important general points do not specifically differentiate between the two cohorts.
How was the hepatic artery anastomosis done? What suture material? Knots continuous or interrupted?
Minor points
‘Microsurgeons should perform hepatic artery anastomosis proficiently with the instruments most proficient’- this sentence in the abstract can be changed to avoid proficient twice
Page 2, line 48- ‘It is a novel high-definition digital camera……from conventional microscopy to exoscope is shorten’. This sentence is long with errors of grammar and syntax. Would suggest reducing it into two shorter sentences which convey the same meaning
Page 2, line 62 ‘collected data included..
Page 2, line 72 control group and study group can be described in past tense
Page 2, line 75 did not provide ‘any’ instead of ‘no’ clinical significance
Page 2, line 79 ‘ were not at all significant difference in favorable of hepatic artery anastomosis’.. sentence grammar can be changed
Reviewer 4 Report
I read with interest the article by Ching-Min et al. regarding Using 3D microscope for hepatic artery reconstruction in living 2 donor liver transplant
The article covers an important topic such as the technical challenge of arterial anastomosis in liver living donation.
However, there are major issues.
What kind of statistical analysis did you do? We don’t have any detail. What kind of outcome did you evaluate? The main limitation of the study is that it described only the technical aspect of arterial anastomosis, without going deeper into the possible other factors that could be correlated with arterial complications. Donors and recipients’ demographics, previous treatments (TACE), MELD score, indications of transplantation, GRWR, cold ischemia time, warm ischemia time, operational time, duration of the arterial reconstruction, the use of conduits, biliary reconstruction (hepaticojejunostomy) are all factors that can be related to arterial complications, as showed by numerous studies (10.3748/wjg.v20.i30.10545; 10.1002/lt.20566; 10.1016/j.jamcollsurg.2008.12.032; 10.1097/TP.0000000000001451). None of these factors are considered and evaluated in the paper. I think that before concluding, as stated in the paper, that “our comparative analysis of hepatic artery reconstruction done via operational microscopy and via 3D digital microscopy showed no statistical nor clinical significance in operative outcome” this topic has to be addressed, as the causes of arterial complications are multifactorial and not describe in the paper.
Another point:
Quote: “The use of exoscopes, such as 3D microscopy, can neutralize mal-posture and alleviate work-related injuries”. How did you prove it? It should be proven with some questionary.
The topic is very interesting, as there is debate about which is the best tool to improve arterial anastomosis in LDLT. I strongly suggest improving the paper with additional details.
Round 2
Reviewer 1 Report
Line 30: Delete "most" before they
Reviewer 4 Report
Thank you for answering the comments. The paper is now improved, but I think is still not suitable to be published as an original article. I really think that more data about the outcome and complications are still essential, especially when you try to evaluate the effect of a new device.